# Self-Distillation as Instance-Specific Label Smoothing

**Zhilu Zhang**
Cornell University
zz452@cornell.edu

**Mert R. Sabuncu**
Cornell Univerisity
msabuncu@cornell.edu

## Abstract

It has been recently demonstrated that multi-generational self-distillation can improve generalization [11]. Despite this intriguing observation, reasons for the enhancement remain poorly understood. In this paper, we first demonstrate experimentally that the improved performance of multi-generational self-distillation is in part associated with the increasing diversity in teacher predictions. With this in mind, we offer a new interpretation for teacher-student training as amortized MAP estimation, such that teacher predictions enable instance-specific regularization. Our framework allows us to theoretically relate self-distillation to label smoothing, a commonly used technique that regularizes predictive uncertainty, and suggests the importance of predictive diversity in addition to predictive uncertainty. We present experimental results using multiple datasets and neural network architectures that, overall, demonstrate the utility of predictive diversity. Finally, we propose a novel instance-specific label smoothing technique that promotes predictive diversity without the need for a separately trained teacher model. We provide an empirical evaluation of the proposed method, which, we find, often outperforms classical label smoothing.

## 1 Introduction

First introduced as a simple method to compress high-capacity neural networks into a low-capacity counterpart for computational efficiency, knowledge distillation [15] has since gained much popularity across various application domains ranging from computer vision to natural language processing [19, 22, 28, 39, 41] as an effective method to transfer knowledge or features learned from a teacher network to a student network. This empirical success is often justified with the intuition that deeper teacher networks learn better representation with greater model complexity, and the "dark knowledge" that teacher networks provide facilitates student networks to learn better representations and hence enhanced generalization performance. Nevertheless, it still remains an open question as to how exactly student networks benefit from this dark knowledge. The problem is made further puzzling by the recent observation that even self-distillation, a special case of the teacher-student training framework in which the teacher and student networks have identical architectures, can lead to better generalization performance [11]. It was also demonstrated that repeated self-distillation process with multiple generations can further improve classification accuracy.

In this work, we aim to shed some light on self-distillation. We start off by revisiting the multi-generational self-distillation strategy, and experimentally demonstrate that the performance improvement observed in multi-generational self-distillation is correlated with increasing diversity in teacher predictions. Inspired by this, we view self-distillation as instance-specific regularization on the neural network softmax outputs, and cast the teacher-student training procedure as performing amortized maximum a posteriori (MAP) estimation of the softmax probability outputs. The proposed framework provides us with a new interpretation of the teacher predictions as instance-specific priors conditioned on the inputs. This interpretation allows us to theoretically relate distillation to label smoothing, a commonly used technique to regularize predictive uncertainty of NNs, and suggests

that regularization on the softmax probability simplex space in addition to the regularization on predictive uncertainty can be the key to better generalization. To verify the claim, we systematically design experiments to compare teacher-student training against label smoothing. Lastly, to further demonstrate the potential gain from regularization on the probability simplex space, we also design a new regularization procedure based on label smoothing that we term "Beta smoothing."

Our contributions can be summarized as follows:

1. We provide a plausible explanation for recent findings on multi-generational self-distillation.
2. We offer an amortized MAP interpretation of the teacher-student training strategy.
3. We attribute the success of distillation to regularization on both the label space and the softmax probability simplex space, and verify the importance of the latter with systematically designed experiments on several benchmark datasets.
4. We propose a new regularization technique termed "Beta smoothing" that improves upon classical label smoothing at little extra cost.
5. We demonstrate self-distillation can improve calibration.

## 2   Related Works

Knowledge distillation was first proposed as a way for model compression [1, 5, 15]. In addition to the standard approach in which the student model is trained to match the teacher predictions, numerous other objectives have been explored for enhanced distillation performance. For instance, distilling knowledge from intermediate hidden layers were found to be beneficial [14, 17, 18, 32, 34, 39]. Recently, data-free distillation, a novel scenario in which the original data for the teacher is unavailable to students, has also been extensively studied [6, 7, 25, 40].

The original knowledge distillation technique for neural networks [15] has stimulated a flurry of interest in the topic, with a large number of published improvements and applications. For instance, prior works [2, 33] have proposed Bayesian techniques in which distributions are distilled with Monte Carlo samples into more compact models like a neural network. More recently, there has also been work on the importance of distillation from an ensemble of model [24], which provides a complementary view on the role of predictive diversity. Lopez-Paz et al. [23] combined distillation with the theory of privileged information, and offered a generalized framework for distillation. To simplify distillation, Zhu et al. [45] proposed a method for one-stage online distillation. There have also been successful applications of distillation for adversarial robustness [28].

Several papers have attempted to study the effect of distillation training on student models. Furlanello et al. [11] examined the effect of distillation by comparing the gradients of the distillation loss against that of the standard cross-entropy loss with ground truth labels. Phuong et al. [31] considered a special case of distillation using linear and deep linear classifiers, and theoretically analyzed the effect of distillation on student models. Cho and Hariharan [8] conducted a thorough experimental analysis of knowledge distillation, and observed that larger models may not be better teachers. Another experimentally driven work to understand the effect of distillation was also done in the context of natural language processing [44]. Most similar to our work is [42], in which the authors also established a connection between label smoothing and distillation. However, our argument comes from a different theoretical perspective and offers complementary insights. Specifically, [42] does not highlight the importance of instance-specific regularization. We also provide a general MAP framework and a careful empirical comparison of label smoothing and self-distillation.

## 3   Preliminaries

We consider the problem of $k$-class classification. Let $\mathcal{X} \subseteq \mathbb{R}^d$ be the feature space and $\mathcal{Y} = \{1, .., k\}$ be the label space. Given a dataset $\mathcal{D} = \{\boldsymbol{x}_i, y_i\}_{n=1}^n$ where each feature-label pair $(\boldsymbol{x}_i, y_i) \in \mathcal{X} \times \mathcal{Y}$, and we are interested in finding a function that maps input features to corresponding labels $f : \mathcal{X} \to \mathbb{R}^c$. In this work, we restrict the function class to the set of neural networks $f_{\boldsymbol{w}}(\boldsymbol{x})$ where $\boldsymbol{w} = \{W_i\}_{i=1}^L$ are the parameters of a neural network with $L$ layers. We define a likelihood model $p(y|\boldsymbol{x}; \boldsymbol{w}) = \mathrm{Cat}\left(\mathrm{softmax}\left(f_{\boldsymbol{w}}(\boldsymbol{x})\right)\right)$, a categorical distribution with parameters $\mathrm{softmax}\left(f_{\boldsymbol{w}}(\boldsymbol{x})\right) \in \Delta(L)$. Here $\Delta(L)$ denotes the $L$-dimensional probability simplex. Typically,

maximum likelihood estimation (MLE) is performed. This leads to the cross-entropy loss

$$\mathcal{L}_{cce}(\boldsymbol{w}) = -\sum_{i=1}^{n}\sum_{j=1}^{k} \boldsymbol{y}_{ij} \log p(y=j|\boldsymbol{x}_i;\boldsymbol{w}), \tag{1}$$

where $\boldsymbol{y}_{ij}$ corresponds to the $j$-th element of the one-hot encoded label $y_i$.

### 3.1 Teacher-Student Training Objective

Given a pre-trained model (teacher) $f_{\boldsymbol{w}_t}$, distillation loss can be defined as:

$$\mathcal{L}_{dist}(\boldsymbol{w}) = -\sum_{i=1}^{n}\sum_{j=1}^{k}[\text{softmax}\big(f_{\boldsymbol{w}_t}(\boldsymbol{x})/T\big)]_j \log p(y=j|\boldsymbol{x}_i;\boldsymbol{w}), \tag{2}$$

where $[\cdot]_j$ denotes the $j$'th element of a vector. A second network (student) $f_{\boldsymbol{w}}$ can then be trained with the following total loss:

$$\mathcal{L}(\boldsymbol{w}) = \alpha\mathcal{L}_{cce}(\boldsymbol{w}) + (1-\alpha)\mathcal{L}_{dist}(\boldsymbol{w}), \tag{3}$$

where $\alpha \in [0,1]$ is a hyper-parameter, and $T$ corresponds to the temperature scaling hyper-parameter that flattens teacher predictions. In self-distillation, both teacher and student models have the same network architecture. In the original self-distillation experiments conducted by Furlanello et al. [11], $\alpha$ and $T$ are set to 0 and 1, respectively throughout the entire training process.

Note that, temperature scaling has been applied differently compared to previous literature on distillation [15]. As addressed in Section 5, we only apply temperature scaling to teacher predictions in computing distillation loss. We empirically observe that this yields results consistent with previous reports. Moreover, as we show in the Appendix A.2, performing temperature scaling only on the teacher but not the student models can lead to significantly more calibrated predictions.

## 4 Multi-Generation Self-Distillation: A Close Look

Self-distillation can be repeated iteratively such that during training of the $i$-th generation, the model obtained at $(i-1)$-th generation is used as the teacher model. This approach is referred to as multi-generational self-distillation, or "Born-Again Networks" (BAN). Empirically it has been observed that student predictions can consistently improve with each generation. However, the mechanism behind this improvement has remained elusive. In this work, we argue that the main attribute that leads to better performance is the increasing uncertainty and diversity in teacher predictions. Similar observations that more "tolerant" teacher predictions lead to better students were also made by Yang et al. [38]. Indeed, due the monotonicity and convexity of the negative log likelihood function, since the element that corresponds to the true label class of the softmax output $p(y=y_i|\boldsymbol{x}_i;\boldsymbol{w})$ is often much greater than that of the other classes, together with early stopping, each subsequent model will likely have increasingly unconfident softmax outputs corresponding to the true label class.

### 4.1 Predictive Uncertainty

We use Shannon Entropy to quantify the uncertainty in instance-specific teacher predictions $p(y|\boldsymbol{x};\boldsymbol{w}_i)$, averaged over the training set, which we call "Average Predictive Uncertainty," and define as:

$$\mathbb{E}_{\boldsymbol{x}}\left[H\left(p(\cdot|\boldsymbol{x};\boldsymbol{w}_i)\right)\right] \approx \frac{1}{n}\sum_{j=1}^{n} H\left(p(\cdot|\boldsymbol{x}_j;\boldsymbol{w}_i)\right) = \frac{1}{n}\sum_{j=1}^{n}\sum_{c=1}^{k} -p(y_c|\boldsymbol{x}_j;\boldsymbol{w}_i)\log p(y_c|\boldsymbol{x}_j;\boldsymbol{w}_i). \tag{4}$$

Note that previous literature [10, 29] has also proposed to use the above measure as a regularizer to prevent over-confident predictions. Label smoothing [29, 35] is a closely related technique that also penalizes over-confident predictions by explicitly smoothing out ground-truth labels. A detailed discussion on the relationship between the two can be found in Appendix A.1.

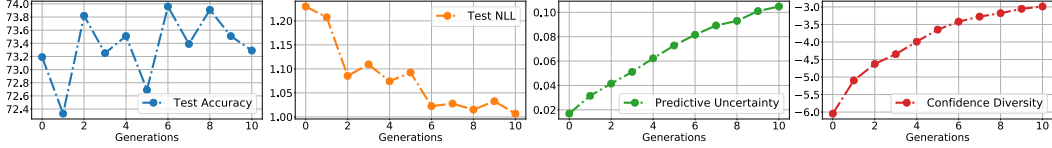

Figure 1: Results for sequential self-distillation over 10 generations are shown above. Model obtained at the $(i-1)$-th generation is used as the teacher model for training at the $i$-th generation. Accuracy and NLL are obtained on the test set using the student model, whereas the predictive uncertainty and confidence diversity are evaluated on the training set with teacher predictions.

## 4.2 Confidence Diversity

Average Predictive Uncertainty is insufficient to fully capture the variability associated with teacher predictions. In this paper, we argue it is also important to consider the amount of spreading of teacher predictions over the probability simplex among different (training) samples. For instance, two teachers can have very similar Average Predictive Uncertainty values, but drastically different amounts of spread on the probability simplex if the softmax predictions of one teacher are much more diverse among different samples than the other. We coin this population spread in predictive probabilities "Confidence Diversity." As we show below, characterizing the Confidence Diversity can be important for understanding teacher-student training.

The differential entropy[1] over the entire probability simplex is a natural measure to quantify the confidence diversity. However, accurate entropy estimation can be challenging, and its computation is severely hampered by the curse of dimensionality, particularly in applications with a large number of classes. To alleviate the problem, in this paper, we propose to measure only the entropy of the softmax element corresponding to the true label class, thereby simplifying the measure to a one-dimensional entropy estimation task. Mathematically, if we denote $c = \phi(\boldsymbol{x}, y) := [\text{softmax}(f_{\boldsymbol{w}}(\boldsymbol{x}))]_y$, and let $p_C$ be the probability density function of the random variable $C := \phi(\boldsymbol{X}, Y)$ where $(\boldsymbol{X}, Y) \sim p(\boldsymbol{x}, y)$, then, we quantify Confidence Diversity via the differential entropy of $C$:

$$h(C) = -\int p_C(c) \log p_C(c)\, dc. \tag{5}$$

We use the KNN-based entropy estimator to compute $h(C)$ over the training set [4]. In essence, the above measure quantifies the amount of spread associated with the teacher predictions on the true label class. The smaller the value, the more similar the softmax values are across different samples.

## 4.3 Sequential Self-Distillation Experiment

We perform sequential self-distillation with ResNet-34 on the CIFAR-100 dataset for 10 generations. At each generation, we train the neural networks for 150 epochs using the identical optimization procedure as in the original ResNet paper [13]. Following Furlanello et al. [11], $\alpha$ and $T$ are set to 0 and 1 respectively throughout the entire training process. Additional experiments with different values of $T$ can be found in Appendix A.3. Fig. 1 summarizes the results. As indicated by the general increasing trend in test accuracy, sequential distillation indeed leads to improvements. The entropy plots also support the hypothesis that subsequent generations exhibit increasing diversity and uncertainty in predictions. Despite the same increasing trend, the two entropy metrics quantify different things. The increase in average predictive uncertainty suggests overall a drop in the confidence of the categorical distribution, while the growth in confidence diversity suggests an increasing variability in teacher predictions. Interestingly, we also see obvious improvements in terms of NLL, suggesting in addition that BAN can improve calibration of predictions [12].

To further study the apparent correlation between student performance and entropy of teacher predictions over generations, we conduct a new experiment, where we instead train a single teacher. This teacher is then used to train a single generation of students while varying the temperature hyperparameter $T$ in Eq. 3, which explicitly adjusts the uncertainty and diversity of teacher predictions. For consistency, we keep $\alpha = 0$. Results are illustrated in Fig. 2. As expected, increasing $T$ leads to

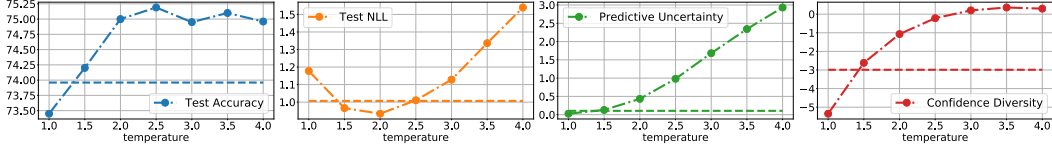

Figure 2: Results with teacher predictions scaled by varying temperature $T$. The flat lines in the plots correspond to the largest/smallest values achieved over 10 generations of sequential distillation with $T = 1$ in the previous experiments for accuracy, predictive uncertainty and confidence diversity/NLL.

greater predictive uncertainty and diversity in teacher predictions. Importantly, we see this increase leads to drastic improvements in the test accuracy of students. In fact, the gain is much greater than the best achieved with 10 generations of BAN with $T = 1$ (indicated with the flat line in the plot). The identified correlation is consistent with the recent finding that early-stopped models, which typically have much larger entropy than fully trained ones, serve as better teachers [8]. Lastly, we also see improvements in NLL with increasing entropy of teacher predictions. However, too high $T$ leads to a subsequent increase in NLL, likely due to teacher predictions that lack in confidence.

A closer look at the entropy metrics of the above experiment reveals an important insight. While the average predictive uncertainty is strictly increasing with $T$, the confidence diversity plateaus after $T = 2.5$. The plateau of confidence diversity coincides closely with the stagnation of student test accuracy, hinting at the importance of confidence diversity in teacher predictions. The apparent correlation between accuracy and confidence diversity can be also seen from the additional sequential self-distillation experiments found in Appendix A.3. This makes intuitive sense. Given a training set, we would expect that some of the samples be much more typically representative of the label class than others. Ideally, we would hope to classify the typical examples with much greater confidence than an ambiguous example of the same class. Previous results show that training with such instance-specific uncertainty can indeed lead to better performance [30]. Our view is that in self-distillation, the teacher provides the means for instance-specific regularization.

## 5 An Amortized MAP Perspective of Self-Distillation

The instance-specific regularization perspective on self-distillation motivates us to recast the training procedure as performing Maximum a posteriori (MAP) estimation on the softmax probability vector. Specifically, suppose now that the likelihood $p(y|\boldsymbol{x}, \boldsymbol{z}) = \text{Cat}(\boldsymbol{z})$ be a categorical distribution with parameter $\boldsymbol{z} \in \Delta(L)$ and the conditional prior $p(\boldsymbol{z}|\boldsymbol{x}) = \text{Dir}(\boldsymbol{\alpha_x})$ be a Dirichlet distribution with instance-specific parameter $\boldsymbol{\alpha_x}$. Due to conjugacy of the Dirichlet prior, a closed-form solution of $\hat{\boldsymbol{z}}_i = \frac{\boldsymbol{c}_i + \boldsymbol{\alpha}_{\boldsymbol{x}_i} - 1}{\sum_j \boldsymbol{c}_j + \boldsymbol{\alpha}_{\boldsymbol{x}j} - 1}$, where $\boldsymbol{c}_i$ corresponds to number of occurrences of the $i$-th category, can be easily obtained.

The above framework is not useful for classification when given a new sample $\boldsymbol{x}$ without any observations $y$. Moreover, in the common supervised learning setup, only one observation of label $y$ is available for each sample $\boldsymbol{x}$. The MAP solution shown above merely relies on the provided label $y$ for each sample $\boldsymbol{x}$, without exploiting the potential similarities among different samples $(\boldsymbol{x}_i)$'s in the entire dataset for more accurate estimation. For example, we could have different samples that are almost duplicates (cf. [3]), but have different $y_i$'s, which could inform us about other labels that could be drawn from $\boldsymbol{z}_i$. Thus, instead of relying on the instance-level closed-form solution, we can train a (student) network to amortize the MAP estimation $\hat{\boldsymbol{z}}_i \approx \text{softmax}\big(f_{\boldsymbol{w}}(\boldsymbol{x}_i)\big)$ with a given training set, resulting in an optimization problem of:

$$\max_{\boldsymbol{w}} \sum_{i=1}^{n} \log p(\boldsymbol{z}|\boldsymbol{x}_i, y_i; \boldsymbol{w}, \boldsymbol{\alpha_x}) = \max_{\boldsymbol{w}} \sum_{i=1}^{n} \log p(y = y_i|\boldsymbol{z}, \boldsymbol{x}_i; \boldsymbol{w}) + \log p(\boldsymbol{z}|\boldsymbol{x}_i; \boldsymbol{w}, \boldsymbol{\alpha_x})$$

$$= \max_{\boldsymbol{w}} \underbrace{\sum_{i=1}^{n} \log[\text{softmax}\,(f_{\boldsymbol{w}}(\boldsymbol{x}_i))]_{y_i}}_{\text{Cross entropy}} + \underbrace{\sum_{i=1}^{n} \sum_{c=1}^{k} ([\boldsymbol{\alpha}_{\boldsymbol{x}_i}]_c - 1) \log[\boldsymbol{z}]_c}_{\text{Instance-specific regularization}}.$$

$$(6)$$

Eq. 6 is an objective that provides us with a function to obtain a MAP solution of $\boldsymbol{z}$ given an input sample $\boldsymbol{x}$. Note that, we do not make any assumptions about the availability or number of label observations of $y$ for each sample $\boldsymbol{x}$. This enables us to find an approximate MAP solution to $\boldsymbol{x}$ at test-time when $\boldsymbol{\alpha_x}$ and $y$ are unavailable. The resulting framework can be generally applicable to various scenarios like semi-supervised learning or learning from multiple labels per sample. Nevertheless, in the following, we restrict our attention to supervised learning with a single label per training sample.

## 5.1 Label Smoothing as MAP

The difficulty now lies in obtaining the instance-specific prior $\mathrm{Dir}(\boldsymbol{\alpha_x})$. A naive independence assumption that $p(\boldsymbol{z}|\boldsymbol{x}) = p(\boldsymbol{z})$ can be made. Under such an assumption, a sensible choice of prior would be a uniform distribution across all possible labels. Choosing $[\boldsymbol{\alpha_x}]_c = [\boldsymbol{\alpha}]_c = \frac{\beta}{k} + 1$ for all $c \in \{1, ..., k\}$ for some hyper-parameter $\beta$, the MAP objective becomes

$$\mathcal{L}_{LS} = \sum_{i=1}^{n} -\log[\boldsymbol{z}]_{y_i} + \beta \sum_{i=1}^{n} \sum_{c=1}^{k} -\frac{1}{k} \log[\boldsymbol{z}]_c. \tag{7}$$

As noted in prior work, this loss function is equivalent to the commonly used label smoothing (LS) regularization [29, 35] (derivations can be found in Appendix A.1). Observe also that the training objective in essence promotes predictions with larger predictive uncertainty, but not confidence diversity.

## 5.2 Self-Distillation as MAP

A better instance-specific prior distribution can be obtained using a pre-trained (teacher) neural network. Let us consider a network $f_{\boldsymbol{w}_t}$ trained with the regular MLE objective, by maximizing $p(y|\boldsymbol{x}; \boldsymbol{w}_t) = \mathrm{Cat}(\mathrm{softmax}(f_{\boldsymbol{w}_t}(\boldsymbol{x}))$, where $[\mathrm{softmax}(f_{\boldsymbol{w}_t}(\boldsymbol{x}))]_i = \frac{[\exp(f_{\boldsymbol{w}_t}(\boldsymbol{x}))]_i}{\sum_j [\exp(f_{\boldsymbol{w}_t}(\boldsymbol{x}))]_j}$. Now, due to conjugacy of the Dirichlet prior, the marginal likelihood $p(y|\boldsymbol{x}; \boldsymbol{\alpha_x})$ is a Dirichlet-multinomial distribution [26]. In the case of single label observation considered, the marginal likelihood reduces to a categorical distribution. As such, we have: $p(y|\boldsymbol{x}; \boldsymbol{\alpha_x}) = \mathrm{Cat}(\overline{\boldsymbol{\alpha_x}})$, where $\overline{\boldsymbol{\alpha_x}}$ is normalized such that $[\overline{\boldsymbol{\alpha_x}}]_i = \frac{[\boldsymbol{\alpha_x}]_i}{\sum_j [\boldsymbol{\alpha_x}]_j}$. We can thus interpret $\exp(f_{\boldsymbol{w}_t}(\boldsymbol{x}))$ as the parameters of the Dirichlet distribution to obtain a useful instance-specific prior on $\boldsymbol{z}$. However, we observe that there is a scale ambiguity that needs resolving, since any of the following will yield the same $\overline{\boldsymbol{\alpha_x}}$:

$$\boldsymbol{\alpha_x} = \beta \exp(f_{\boldsymbol{w}_t}(\boldsymbol{x})/T) + \gamma, \tag{8}$$

where $T = 1$ and $\gamma = 0$, and $\beta$ corresponds to some hyper-parameter. Using $T > 1$ and $\gamma > 0$ corresponds to flattening the prior distribution, which we found to be useful in practice - an observation consistent with prior work. Note that in the limit of $T \to \infty$, the instance-specific prior reduces to a uniform prior corresponding to classical label smoothing. Setting $\gamma = 1$ (we also experimentally explore the effect of varying $\gamma$. See Appendix A.10 for details), we obtain

$$\boldsymbol{\alpha_x} = \beta \exp(f_{\boldsymbol{w}_t}(\boldsymbol{x})/T) + 1 = \beta \sum_j [\exp(f_{\boldsymbol{w}_t}(\boldsymbol{x})/T)]_j \, \mathrm{softmax}(f_{\boldsymbol{w}_t}(\boldsymbol{x})/T) + 1. \tag{9}$$

Plugging this into Eq. 6 yields

$$\mathcal{L}_{SD} = \sum_{i=1}^{n} -\log[\boldsymbol{z}]_{y_i} + \beta \sum_{i=1}^{n} \omega_{\boldsymbol{x}_i} \sum_{c=1}^{k} -[\mathrm{softmax}(f_{\boldsymbol{w}_t}(\boldsymbol{x}_i)/T)]_c \log[\boldsymbol{z}]_c, \tag{10}$$

very similar to the distillation loss of Eq. 3, with an additional sample-specific weighting term $\omega_{\boldsymbol{x}_i} = \sum_j [\exp(f_{\boldsymbol{w}_t}(\boldsymbol{x}_i)/T)]_j$!

Despite the interesting result, we empirically observe that, with temperature values $T$ found to be useful in practice, the relative weightings of samples are too close to yield a significant difference from regular distillation loss. Hence, for all of our experiments, we still adopt the distillation loss of Eq. 3. However, we believe that, with teacher models trained with an objective more appropriate than MLE, the difference might be bigger. We hope to explore alternative ways of obtaining teacher models to effectively utilize the sample re-weighted distillation objective as future work.

The MAP interpretation, together with empirical experiments conducted in Section 4, suggests that multi-generational self-distillation can in fact be seen as an inefficient approach to implicitly flatten and diversify the instance-specific prior distribution. Our experiments suggest that instead, we can more effectively tune for hyper-parameters $T$ and $\gamma$ to achieve similar, if not better, results. Moreover, from this perspective, distillation in general can be understood as a regularization strategy. Some empirical evidence for this can be found in Appendix A.6 and A.7.

### 5.3 On the Relationship between Label Smoothing and Self-Distillation

The MAP perspective reveals an intimate relationship between self-distillation and label smoothing. Label smoothing increases the uncertainty of predictive probabilities. However, as discussed in Section 4, this might not be enough to prevent overfitting, as evidenced by the stagnant test accuracy despite increasing uncertainty in Fig. 2. Indeed, the MAP perspective suggests that, ideally, each sample should have a distinct probabilistic label. Instance-specific regularization can encourage confidence diversity, in addition to predictive uncertainty.

While the predictive uncertainty can be explicitly used for regularization as previously discussed in Section 4.1, we observe empirically that promoting confidence diversity directly through the proposed measure in Section 4.2 can be hard in practice, yielding unsatisfactory results. This could have been caused by difficulty in estimating confidence diversity accurately using mini-batch samples. Naively promoting confidence diversity during the early stage of training could also have harmed learning. As such, we can view distillation as an indirect way of achieving this objective. We leave it as future work to further explore alternative techniques to enable direct regularization of confidence diversity.

## 6 Beta Smoothing Labels

Self-distillation requires training a separate teacher model. In this paper, we propose an efficient enhancement to label smoothing strategy where the amount of smoothing will be proportional to the uncertainty of predictions. Specifically, we make use of the exponential moving average (EMA) predictions as implemented by Tarvainen and Valpola [36] of the model at training, and obtain a ranking based on the confidence (the magnitude of the largest element of the softmax) of predictions at each mini-batch, on the fly, from smallest to largest. Instead of assigning uniform distributions $[\boldsymbol{\alpha_x}]_c = \frac{\beta}{k} + 1$ for all $c \in \{1, ..., k\}$ to all samples as priors, during each iteration, we sample and sort a set of i.i.d. random variables $\{b_1 \leq ... \leq b_m\}$ from $Beta(a, 1)$ where $m$ corresponds to the mini-batch size and $a$ corresponds to the hyper-parameter associated with the Beta distribution. Then, we assign $[\boldsymbol{\alpha_{x_i}}]_{y_i} = \beta b_i + 1$ and $[\boldsymbol{\alpha_{x_i}}]_c = \beta \frac{1-b_i}{k-1} + 1$ for all $c \neq y_i$ as the prior to each sample $\boldsymbol{x}_i$, based on the ranking obtained. In this way, samples with larger confidence obtained through the EMA predictions will receive less amount of label smoothing and vice versa. Thus, the amount of label smoothing applied to a sample will be proportional to the amount of confidence the model has about that sample's prediction. Those instances that are more challenging to classify will, therefore, have more smoothing applied to their labels.

In practice, for consistency with distillation, Eq. 3 is used for training. Beta-smoothed labels of $b_i$ on the ground truth class and $\frac{1-b_i}{k-1}$ on all other classes are used in lieu of teacher predictions for each $\boldsymbol{x}_i$. Lastly, note that EMA predictions are used in order to stabilize the ranking obtained at each iteration of training. We empirically observe a significant performance boost with the EMA predictions. We term this method *Beta smoothing*.

To better examine the role of EMA predictions has on Beta smoothing, we conduct two ablation studies. Firstly, since the EMA predictions are used for Beta smoothed labels, we compare the effectiveness of Beta smoothing against self-training explicitly using the EMA predictions (see Appendix A.5 for details). Moreover, to test the importance of ranking obtained from EMA predictions, we include in the Appendix A.8 an additional experiment for which random Beta smoothing is applied to each sample.

Beta smoothing regularization implements an instance-specific prior that encourages confidence diversity, and yet does not require the expensive step of training a separate teacher model. We note that, due to the constantly changing prior used at every iteration of training, Beta smoothing does not, strictly speaking, correspond to the MAP estimation in Eq. 6. Nevertheless, it is a simple and effective way to implement the instance-specific prior strategy. As we demonstrate in the following section,

it can lead to much better performance than label smoothing. Moreover, unlike teacher predictions which have unique softmax values for all classes, the difference between Beta and label smoothing only comes from the ground-truth softmax element. This enables us to conduct more systematic experiments to illustrate the additional gain from promoting confidence diversity.

# 7 Empirical Comparison of Distillation and Label Smoothing

To further demonstrate the benefits of the additional regularization on the softmax probability vector space, we design a systematic experiment to compare self-distillation against label smoothing. In addition, experiments on Beta smoothing are also conducted to further verify the importance of confidence diversity, and to promote Beta smoothing as a simple alternative that can lead to better performance than label smoothing at little extra cost. We note that, while previous works have highlighted the similarity between distillation and label smoothing from another perspective [42], we provide a detailed empirical analysis that uncovers additional benefits of instance-specific regularization.

## 7.1 Experimental Setup

We conduct experiments on CIFAR-100 [20], CUB-200 [37] and Tiny-imagenet [9] using ResNet [13] and DenseNet [16]. We follow the original optimization configurations, and train the ResNet models for 150 epochs and DeseNet models for 200 epochs. 10% of the training data is split as the validation set. All experiments are repeated 5 times with random initialization. For simplicity, label smoothing is implemented with explicit soft labels instead of the objective in Eq. 7. We fix $\epsilon = 0.15$ in label smoothing for all our experiments (additional experiments with $\epsilon = 0.1, 0.3$ can be found in the Appendix A.4). The hyper-parameter $\alpha$ of Eq. 3 is taken to be 0.6 for self-distillation. Only one generation of distillation is performed for all experiments. To systematically decompose the effect of the two regularizations in self-distillation, given a pre-trained teacher and $\alpha$, we manually search for temperature $T$ such that the average effective label of the ground-truth class, $\alpha + (1 - \alpha)[\text{softmax}\,(f_{\boldsymbol{w}_t}(\boldsymbol{x}_i)/T)]_{y_i}$, is approximately equal to 0.85 to match the hyper-parameter $\epsilon$ chosen for label smoothing. Eq. 3 is also used for Beta smoothing with $\alpha = 0.4$. The parameter $a$ of the Beta distribution is set such that $\mathbb{E}[\alpha + (1 - \alpha)b_i] = \epsilon$, to make the average probability of ground truth class the same as $\epsilon-$label smoothing.

We emphasize that the goal of the experiment is to methodically decompose the gain from the two aforementioned regularizations of distillation. Note that, both $\alpha$ and $T$ can influence the amount of predictive uncertainty and confidence diversity in teacher predictions at the same time. This coupled effect can make hyper-parameter tuning hard. Due to limited computational resources, hyper-parameter tuning is not performed, and the results for all methods can be potentially enhanced. Lastly, we also incorporate an additional distillation experiment in which the deeper DenseNet model is used as the teacher model for comparison against self-distillation. Results can be found in Appendix A.9.

## 7.2 Results

Test accuracies are summarized in the top row for each experiment in Fig. 3. Firstly, all regularization techniques lead to improved accuracy compared to the baseline model trained with cross-entropy loss. In agreement with previous results, self-distillation performs better than label smoothing in all of the experiments with our setup, in which the effective degree of label smoothing in distillation is, on average, the same as that of regular label smoothing. The results suggest the importance of confidence diversity in addition to predictive uncertainty. It is worth noting that we obtain encouraging results with Beta smoothing. Outperforming label smoothing in all but the CIFAR-100 ResNet experiment, it can even achieve comparable performance to that of self-distillation for the CUB-200 dataset with no separate teacher model required. The improvements of Beta smoothing over label smoothing also serve direct evidence on the importance of confidence diversity, as the only difference between the two is the additional spreading of the ground truth classes. We hypothesize that the gap in accuracy between Beta smoothing and self-distillation is mainly due to better instance-specific priors set by a pre-trained teacher network. The differences in the non-ground-truth classes between the two methods could also account for the small gap in accuracy performance.

Results on calibration are shown in the bottom rows of Fig. 3, where we report the expected calibration error (ECE) [12]. As anticipated, all regularization techniques lead to enhanced calibration.

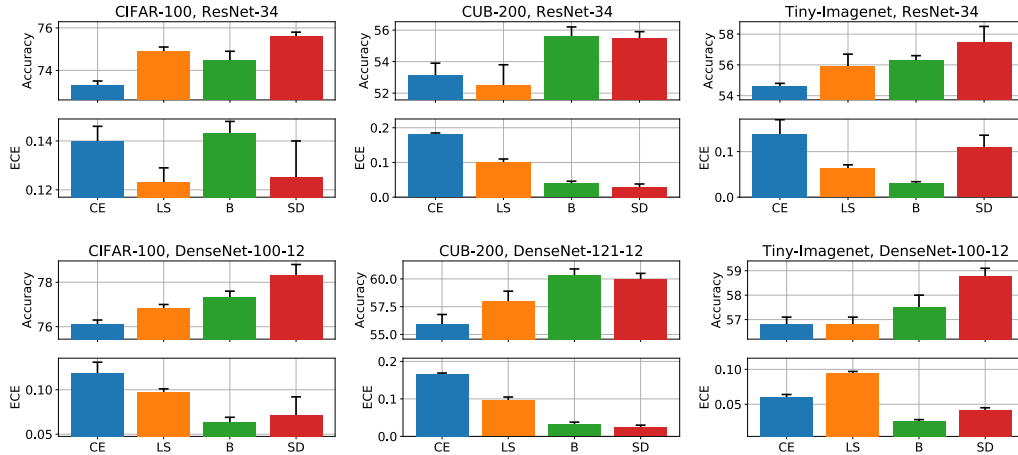

Figure 3: Experimental Results performed on CIFAR-100, CUB-200 and the Tiny-Imagenet dataset. "CE", "LS", "B" and "SD" refers to "Cross Entropy", "Label Smoothing", "Beta Smoothing" and "Self-Distillation" respectively. The top rows of each experiment show bar charts of accuracy on test set for each experiment conducted, while the bottom rows are bar charts of expected calibration error.

Nevertheless, we see that the errors obtained with self-distillation are much smaller in general compared to label smoothing. As such, instance-specific priors can also lead to more calibrated models. Beta smoothing again not only produces models with much more calibrated predictions compared to label smoothing but compares favorably to self-distillation in a majority of the experiments.

# 8 Discussion and Future Directions

In this paper, we provide empirical evidence that diversity in teacher predictions is correlated with student performance in self-distillation. Inspired by this observation, we offer an amortized MAP interpretation of the popular teacher-student training strategy. The novel viewpoint provides us with insights on self-distillation and suggests ways to improve it. For example, encouraged by the results obtained with Beta smoothing, there are possibly better and/or more efficient ways to obtain priors for instance-specific regularization.

Recent literature shows that label smoothing leads to better calibration performance [27]. In this paper, we demonstrate that distillation can also yield more calibrated models. We believe this is a direct consequence of not performing temperature scaling on student models during training. Indeed, with temperature scaling also on the student models, the student logits are likely pushed larger during training, leading to over-confident predictions.

More generally, we have only discussed the teacher-student training strategy as MAP estimation. There have been other recently proposed techniques involving training with soft labels, which we can interpret as encouraging confidence diversity or implementing instance-specific regularization. For instance, the mixup regularization [43] technique creates label diversity by taking random convex combinations of the training data, including the labels. Recently proposed consistency-based semi-supervised learning methods such as [21, 36], on the other hand, utilize predictions on unlabeled training samples as an instance-specific prior. We believe this unifying view of regularization with soft labels can stimulate further ideas on instance-specific regularization.

### Acknowledgments

This work was supported by NIH R01 grants (R01LM012719 and R01AG053949), the NSF NeuroNex grant 1707312, and NSF CAREER grant (1748377).

**Statement of the Potential Broader Impact**

In this paper, we offer a new interpretation of the self-distillation training framework, a commonly used technique for improved accuracy used among practitioners in the deep learning community, which allows us to gain some deeper understanding of the reasons for its success. With the ubiquity of deep learning in our society today and countless potential future applications of it, we believe our work can potentially bring positive impacts in several ways.

Firstly, despite the empirical utility of distillation and numerous successful applications in many tasks and applications ranging from computer vision to natural language processing problems, we still lack a thorough understanding of why it works. In our opinion, blindly applying methods and algorithms without a good grasp on the underlying mechanisms can be dangerous. Our perspective offers a theoretically grounded explanation for its success that allows us to apply the techniques to real-world applications broadly with greater confidence.

In addition, the proposed interpretation of distillation as a regularization to neural networks can potentially allow us to obtain models that are more generalizable and reliable. This is an extremely important aspect of applying deep learning to sensitive domains like healthcare and autonomous driving, in which wrong predictions made by machines can lead to catastrophic consequences. Moreover, our new experimental demonstration that models trained with the distillation process can potentially lead to better-calibrated models that can facilitate safer and more interpretable applications of neural networks. Indeed, for real-world classification tasks like disease diagnosis, in addition to accurate predictions, we need reliable estimates of the level of confidence of the predictions made, which is something that neural networks are lacking currently as pointed out by recent research. More calibrated models, in our opinion, enhances the explainability and transparency of neural network models.

Lastly, we believe the introduced framework can stimulate further research on the regularization of deep learning models for better generalization and thus safer applications. It was recently demonstrated that deep neural networks do not seem to suffer from overfitting. Our finding suggests that overfitting can still occur, though in a different way than conventional wisdom, and deep learning can still benefit from regularization. As such, we encourage research into more efficient and principled forms of regularization to improve upon the distillation strategy.

We acknowledge the risks associated with our work. To be more specific, our finding advocates for the use of priors for the regularization of neural networks. Despite the potentially better generalization performance of trained models, depending on the choice of priors used for training, unwanted bias can be inevitably introduced into the deep learning system, potentially causing issues of fairness and privacy.

## Footnotes

[1]This is distinct from the average predictive uncertainty discussed in the previous section, which measures the average Shannon entropy of probability vectors.

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
