[Supplementary Material]

# A  Appendix

## A.1   On Label Smoothing and Predictive Uncertainty Regularization

We first give a derivation on the equivalence of label smoothing regularization and Eq. 7. With some simple rearrangement of the terms,

$$\mathcal{L}_{LS} = \sum_{i=1}^{n} -\log[\boldsymbol{z}]_{y_i} + \beta \sum_{i=1}^{n} \sum_{c=1}^{k} -\frac{1}{k} \log[\boldsymbol{z}]_c$$

$$= -(1+\beta)\sum_{i=1}^{n}\left(\frac{k+\beta}{k(1+\beta)}\log[\boldsymbol{z}]_{y_i} + \sum_{c\neq y_i}\frac{\beta}{k(1+\beta)}\log[\boldsymbol{z}]_c\right).$$

The above objective is clearly equivalent to the label smoothing regularization with $1 - \epsilon = \frac{k+\beta}{k(1+\beta)}$, up to a constant factor of $(1+\beta)$.

Label smoothing regularizes predictive uncertainty. The amount of regularization is controlled by the amount of smoothing applied. Evidently, the objective does not regularize confidence diversity. Indeed, assuming a NN with capacity capable of fitting the entire training data, predictions on training data will be pushed arbitrarily close to the smoothed soft label. Empirical evidence for this form of overfitting can be seen from experiments done by Müller et al. [27], in which the authors demonstrated that applying label smoothing leads to hampered distillation performance. The authors hypothesize that this is likely due to erasure of "relative information between logits" when label smoothing is applied, hinting at the overfitting of predictions to the smoothed labels.

A closely related regularization technique is to explicitly regularize on predictive uncertainty:

$$\mathcal{L}_{PU} = \sum_{i=1}^{n} -\log[\boldsymbol{z}]_{y_i} + \beta\frac{1}{n}\sum_{j=1}^{n}\sum_{c=1}^{k}[\boldsymbol{z}]_c\log[\boldsymbol{z}]_c.$$

Prior papers [10, 29] have demonstrated that directly regularizing predictive uncertainty can lead to better performance than label smoothing. However, we note that the above objective does not regularize confidence diversity either. In fact, it can be easily solved, with the method of Lagrange multiplier, that the optima for the objective above is achieved when $[\boldsymbol{z}]_{y_i} = \frac{1}{\beta W(\exp(-1/\beta)(k-1)/\beta)+1}$ where $W$ corresponds to the Lambert W function, and $[\boldsymbol{z}]_c = \frac{1-[\boldsymbol{z}]_{y_i}}{k-1}$ for all $c \neq y_i$, for all sample pairs $(\boldsymbol{x}_i, y_i)$. As such, the global optima obtained by directly regularizing predictive uncertainty is identical to that of label smoothing. In practice, differences between the two can arise due to the details of the optimization procedure (like early stopping), and/or due to model capacity.

## A.2    Additional Experiments with Temperature Scaling on Student Models

Figure 4: *Left:* Test accuracies of ResNet-34 models on the CIFAR-100 dataset when varying temperature. *Right:* ECE of ResNet-34 models on the CIFAR-100 dataset when varying temperature. "Scale both" corresponds to the originally proposed distillation objective in which both teacher and student models are temperature-scaled during training. "Scale teacher only" corresponds to only temperature scaling teacher models during distillation. The *green flat line* represents the performance achieved by the teacher model trained with cross-entropy loss.

To examine the effect of not applying temperature scaling on student models, we conduct an experiment to compare models trained with and without temperature scaling on student models for distillation loss with the ResNet-34 on the CIFAR-100 dataset, using the training objective of Eq. 3. On top of the hyper-parameter $\alpha = 0.4$ used for experiments in Section 7, we also include results with $\alpha = 0.1$, a widely used value for knowledge distillation in prior work [8]. We vary the amount of temperature scaling applied to illustrate the effect of different temperatures have on student models.

Plots of test accuracy and ECE against amount of temperature scaling applied are shown in Fig. 4. Firstly, we observe that models trained with student scaling have ECE almost identical to that of the teacher models. As a direct contrast, we see that the student models trained without student scaling perform much better in terms of calibration error in general over its teacher. Note that the relatively large ECE when $\alpha = 0.4$ and $T > 3$ is likely due to overly unconfident teacher predictions. In addition, we highlight that, with the optimal hyper-parameters of $\alpha$ and $T$ used, student models trained without student scaling can also outperform significantly in terms of test accuracy. We acknowledge that there can be conflicts between the performance of ECE and accuracy, as seen from superior test accuracy but poor ECE achieved for $\alpha = 0.4$ and $T = 4.0$. In practice, we can use the negative log likelihood, a metric influenced by both ECE and accuracy, to find the optimal $\alpha$ and $T$. Lastly, we note that, both $\alpha$ and $T$ alter the amount of predictive uncertainty and confidence diversity in teacher predictions at the same time. This coupled effect could be the reason for the observed conflict between ECE and accuracy. We leave it as a future work to explore alternative ways to decouple the two measures for more efficient and effective parameter search. We believe a decoupled set of parameters can lead to models with better calibration and accuracy at the same time.

## A.3 Additional Experiments on Sequential Self-Distillation with Different Temperatures

Figure 5: Results for sequential self-distillation over 5 generations are shown above for different temperatures. *Top:* temperature $T = 2.0$; *Bottom:* temperature $T = 3.0$. The same temperatures are used throughout the entire sequential distillation process. Model obtained at the $(i - 1)$-th generation is used as the teacher model for training at the $i$-th generation. Accuracy and NLL are obtained on the test set using the student model, whereas the predictive uncertainty and confidence diversity are evaluated on the training set with teacher predictions.

To further verify the observation on predictive uncertainty and confidence diversity made empirically in Section 4, we conduct additional sequential self-distillation experiments with different values of temperature. Figure 5 summarizes the results when temperature is 2 (*top*) and 3 (*bottom*) respectively. As seen clearly, test accuracy and NLL performance correlate strongly with that of confidence diversity, further demonstrating the importance of confidence diversity for greater generalizability in neural networks.

## A.4   Additional Experiments with Different Amount of Label Smoothing $\epsilon$

Figure 6: Experimental Results performed on CIFAR-100, CUB-200 and the Tiny-Imagenet dataset with different amount of label smoothing. *Top:* $\epsilon = 0.1$, *Bottom:* $\epsilon = 0.3$. "CE", "LS", "B" and "SD" refers to "Cross Entropy", "Label Smoothing", "Beta Smoothing" and "Self-Distillation" respectively. The top rows of each experiment show bar charts of accuracy on test set for each experiment conducted, while the bottom rows are bar charts of expected calibration error.

In order to verify that the conclusions drawn from our empirical experiments hold more generally, we conduct additional experiments varying the amount of label smoothing $\epsilon$. Additional smoothing parameters of $\epsilon = 0.1$ and $\epsilon = 0.3$ are used. As a fair comparison, given the label smoothing parameter $\epsilon$, hyper-parameters for Beta smoothing and self-distillation are adjusted so that the amount of label smoothing for samples on average is the same as that of label smoothing. Experimental results are summarized in Figure 6. Observe that the general trend in terms of both the accuracy and calibration holds across different values of $\epsilon$.

## A.5   Additional Experiments with Self-Training Using EMA-Predictions

Figure 7: Additional results to compare Beta smoothing against self-training explicitly with the EMA predictions. "B" and "ST" refer to "beta smoothing" and "self-training" respectively. The top rows of each experiment show bar charts of accuracy on the test set for each experiment conducted, while the bottom rows are bar charts of expected calibration error.

The proposed beta smoothing involves the use of EMA predictions to rank the confidence of samples within each minibatch during training in order to achieve instance-specific regularization. To further demonstrate that the gain in accuracy and calibration obtained through beta smoothing mainly comes from instance-specific regularization, we compare Beta smoothing against explicit self-training using the EMA predictions in which the EMA predictions are directly used as soft labels to compute cross-entropy loss. We follow the training procedure as described in [36] for self-training with EMA predictions. Results using ResNet for all the datasets considered in this paper are summarized in Figure 7. Beta smoothing outperforms self-training using EMA predictions on all of the experiments conducted in terms of both accuracy and calibration. As such, while EMA predictions can be used as a reliable proxy to rank the relative confidence of samples, the predictions themselves are sub-optimal when used as teachers directly.

## A.6 Additional Experiments with CIFAR-10 When Varying Trainset Size

Figure 8: *Left:* Test accuracies of ResNet-34 models on the CIFAR-10 dataset for the teacher and student models when the training set size is varied. *Right:* The relative improvements in accuracy when the training set size is varied.

Recent results show relatively small gain when performing knowledge distillation on the CIFAR-10 dataset [8, 11]. Our perspective of distillation as regularization provides a plausible explanation for this observation. Like all other forms of regularization, its effect diminishes with increasing the size of training data. We experimentally verify the claim by training ResNet-34 models with a varying number of training samples. The experiment are repeated 3 times. Fig. 8 summarizes the results. As expected, increasing sample size leads to an increase in test accuracy for both of the models. Nevertheless, the relative improvement in the accuracy of the student model compared to the teacher decreases as the size of the training set increases, indicating that distillation is a form of regularization.

## A.7 Additional Experiments with CIFAR-100 When Varying Weight Decay

Figure 9: *Left:* Test accuracies of ResNet-34 models on the CIFAR-100 dataset for the teacher and student models when the weight decay hyper-parameter is varied. *Right:* The relative improvements in accuracy when the weight decay hyper-parameter is varied.

To further demonstrate that distillation is a regularization process, we also conduct an additional experiment on the CIFAR-100 dataset using ResNet-34, varying only the weight decay hyper-parameter. Intuitively, larger weight decay regularization makes NNs less prone to overfitting, which should, in turn, reduced the additional benefits obtainable from self-distillation, if it is indeed a form of regularization. To keep the quality of priors identical across all student models, we use the same teacher model obtained from using a weight decay of $10^{-4}$ for all distillation. Our results are summarized in Fig. 9. It is evident that increasing the weight decay hyper-parameter leads to much smaller improvement in test accuracy. Interestingly, we see a noticeable gain in accuracy for baselines models trained with cross-entropy when adjusting the weight decay term, contradicting some of the recent findings that weight decay is ineffective for neural networks.

## A.8 Additional Experiments on Beta Smoothing

Figure 10: Ablation study on Beta smoothing. "LS", "RB" and "B" refers to "Label Smoothing", "Random Beta Smoothing" and "Beta Smoothing" respectively. The top rows of each experiment show bar charts of accuracy on the test set for each experiment conducted, while the bottom rows are bar charts of expected calibration error.

We conduct an ablation study on the proposed Beta smoothing regularization in order to demonstrate the importance of relative ranking. To do so, we run experiments with the identical setup as described in Section 7 for Beta smoothing with completely randomly assigned soft label noise from Beta distribution instead. We term this the "random Beta smoothing". Results are shown in Fig. 10. For convenience, we also include results obtained with regular label smoothing as a benchmark comparison. As seen clearly, the proposed Beta smoothing with ranking obtained from EMA predictions leads to much better results in general in terms of both accuracy and ECE, suggesting that naively encouraging confidence diversity does not lead to significant improvements, and the relative confidence among different samples is also an important aspect in order to obtain better student models. This ablation study also serves as indirect evidence for why self-distillation still outperforms Beta smoothing - with a pre-trained model, much more reliable relative confidence among training samples can be obtained.

## A.9 Additional Experiments on the Effect of Quality of Teachers

Figure 11: Additional results on cross-distillation. "SD" and "CD" refers to "self-distillation" and "cross-distillation" respectively. The top rows of each experiment show bar charts of accuracy on the test set for each experiment conducted, while the bottom rows are bar charts of expected calibration error.

We also perform an additional experiment with the identical setup as described in Section 7 on *cross distillation* of the ResNet and the DenseNet models, in which a ResNet-34 teacher is used to train the DenseNet-100 student and vice versa in an attempt to examine the effect of better/worse priors in self-distillation. Hyper-parameters are fixed in this case such that the predictive uncertainty and diversity associated with the label predictions remain the same as that for self-distillation. Results are summarized in Fig. 11. As seen clearly from consistently better/worse performance of cross distillation for ResNet/DenseNet, better teachers lead to better performance. Thus, in addition to diversity among teacher predictions, the quality of the instance-specific prior used is also important for better generalization performance. Lastly, we also see an apparent benefit in terms of model calibration when a better teacher model is used.

The interpretation of distillation as sample-specific regularization provides us with a reasonable explanation of why deeper NNs are potentially better teachers. With greater capacity, deeper networks can learn better representations that capture more closely the true underlining relative confidence among samples, thereby generating better priors and hence better performance. When too expressive models are used, however, there can be so much overfitting to the ground truth labels that the meaningful rankings are destroyed, despite better accuracy. Recent findings experimentally corroborate our argument [8]. Similar observations were also made when label smoothing is applied [27]. From the regularization perspective, distillation can be also applied to very deep networks for potential improvements, and shallower teacher models can also serve as teacher models for deeper student networks.

## A.10 Additional Experiments on Varying $\gamma$

Figure 12: Additional results on pruned distillation. "SD" and "PD" refer to "self-distillation" and "pruned-distillation" respectively. The top rows of each experiment show bar charts of accuracy on the test set for each experiment conducted, while the bottom rows are bar charts of expected calibration error.

In addition, we consider a simple variation to distillation loss by varying $\gamma$. However, directly adjusting $\gamma$ can be problematic in practice. To understand the effect of changing $\gamma$, suppose we have some $\gamma$ such that $[\boldsymbol{\alpha_x}]_c - 1 < 0$ for some $c \in \{1, ..., k\}$. Since the minimization objective with respect to this class is $-([\boldsymbol{\alpha_x}]_c - 1) \log([\boldsymbol{z}]_c)$, the closer the $[\boldsymbol{z}]_c$ to 0, the smaller the loss function. This leads to numerical issues as the overall loss function can be pushed to negative infinity by forcing $[\boldsymbol{z}]_c$ arbitrarily close to zero.

To circumvent the numerical problem during optimization, we make the observation that the above objective is essentially equivalent to setting the particular element with $[\boldsymbol{\alpha_x}]_c - 1 < 0$ to zero. As such, adjusting the threshold $\gamma$ enables us to prune out the smallest elements of the teacher predictions. To further force the pruned elements to zero, a new softmax probability vector is computed with the remaining elements. In practice, setting the optimal $\gamma$ can be challenging. We instead choose to prune out a fixed percentage of classes for all samples. For instance, pruning $50\%$ of the classes for a 100-class classification amounts to using only the top 50 most confident samples to compute softmax and setting the remaining to zero. We term this method the *pruned-distillation*.

We show some preliminary results with pruned-distillation with $50\%$ of the classes pruned during distillation in Fig. 12. While the performance overall remains the same for the CIFAR-100 and Tiny-Imagenet datasets, a slight improvement can be seen for CUB-200 in terms of both the accuracy and ECE, suggesting the method as an easy-to-implement adjustment with no harm.