[Reviews · NeurIPS 2020]

Review 1

Summary and Contributions: This paper provides a new interpretation for teacher-student training as amortized MAP estimation, relates self-distillation to label smoothing regularization, and proposes an instance-specific label smoothing regularization to improve the network generalization ability further. Experiments verify the effectiveness of proposed method to an extent.

Strengths: The idea relating the performance of self-distillation to the proposed confidence diversity seems interesting.

Weaknesses: The relationships between different parts of this paper are not clearly stated. For example, the authors claimed that they theoretically relate self-distillation to label smoothing. But I think they actually not. They do provide some cues, but they neither theoretically prove such relation nor demonstrate sufficient empirically evidences.

Correctness: About correct.

Clarity: Not very well.

Relation to Prior Work: Yes.

Reproducibility: No

Additional Feedback: Some issues need to be addressed: 1) I don't think the success of self-distillation can be attributed to performing label smoothing regularization. They may share some similarities, but they are quite different. And I didn't find the paper provide direct evidences to prove the assumption that self-distillation relates to label smoothing. The authors should provide more clear description and more direct evidences (theoretically or empirically) to show the relationships between confidence diversity, MAP, self-distillation, and label smoothing. 2) Some technical details are not described clearly. E.g. Line 223-232, the details about the correspondence between b and x are not clear. How are the confidences sorted? How are b sorted? They are in the same order or opposite order? Without these details, it is hard to know what the authors actually did. 3) For the experiments, the proposed Beta-Smooting works worse than the self-distillation method. And the performance of label smoothing don't match with the self-distillation. All these evidences seem conflict with what the authors have claimed in the paper. 4) The authors should provide some explanations before using some abbreviations. For example, Line 137, "NLL", "BAN", etc. Some typos exists, e.g. Line 25, enhanced --> enhance; Line 20, various applications domains; Line 105, likely --> be likely to; Line 109, define --> is defined; Line 207, weightings --> weights; were --> are; etc. ------------- After Rebuttal ------------ I agree with the author that self-distillation imposes an instance-specific prior to regularize the training process. And from this perspective, there exist some similarities with label smoothing. But, what I am interested is to how to understand the characteristics of such prior provided by the self-distillation. Just claiming it is a kind of instance-specific prior is not enough. Further, I would like to see if it is possible to explicitly model such prior or if we can improve the self-distillation from such perspective. The authors relate self-distillation to label smoothing and proposes beta-smoothing. But empirically, we see the proposed beta-smoothing works worse than the self-distillation and the performance of label smoothing cannot match with that of self-distillation. In a word, I don't think relating self-distillation to label smoothing or instance-specific prior explains the success of self-distillation in practice. The rebuttal didn't address the above concerns. Thus I would keep my original rating.


Review 2

Summary and Contributions: The paper investigate an interpretation of self-distillation in terms of label smoothing, a mechanism to regularize predictions so as to avoid excessively confident ones. In particular, the paper proposes an interpretation of the role of multi-generational self-distillation as MAP inference of the distribution of teacher predictions, which emphasizes the importance of predictive diversity in addition to only predictive uncertainty (which is the only mechanism implemented by conventional label smoothing).

Strengths: - The paper is trying to elucidate interesting theoretical question with direct practical ramifications in the training neural networks. - The structure of the paper is exemplary as a paper trying to glean scientific understanding of an empirical phenomenon observed in the training of neural networks: a theoretical model of the mechanism under study is developed and presented along with relevant quantities aimed at validating the predictions of the developed theory in a series of empirical evaluations. In addition the authors present a theory-motivated improvement (beta smoothing of the labels) of the mechanism they hypothesize as underlying the phenomenon under study

Weaknesses: - The paper didn't cite a very relevant line of work on model compression (by for instance Caruana et al) but only focuses on the more modern take that goes under the brand of "distillation". Distilling ensembles of teachers given by previous generations of students is in fact a setting that is arguably very related to multi-generational self-distillation. Moreover, the conceptual framework provided by ensemble learning could have given rise to some interesting hypothesis as to how diversity in prediction uncertainty comes about and how it is useful at test time. - The presentation could be slightly improved. In particular, the final regularized loss that is being optimized is never written down explicitly, which hinders a little bit the clarity of the exposition.

Correctness: The main claims are convincingly supported by theoretical analysis, which is also validated convincingly by experiments showing that multi-generational self-distillation increases confidence diversity as a function of student generations.

Clarity: As mentioned earlier, the presentation could be improved, in that the final regularized loss that is being optimized is never written down explicitly. Backtracking a few steps along the sequence of equations it seems that the end result is essentially the addition of a distillation loss term with an "unnormalized" softmax, although I concede that I might have misunderstood this part.

Relation to Prior Work: The connection to the model compression literature is completely absent. In particular, the papers on compressing ensembles of models could have provided a complementary view on the role of predictive diversity and how it comes about in multi-generational self-distillation (since a sequence of students can be thought of as a teacher consisting in an ensemble of models).

Reproducibility: Yes

Additional Feedback: - The notation could be improved a bit. The vectors z for instance could be explicitly indicized with i for instance in equation (6) and (7) (if I understood correctly). - There is a dC missing in equation (5). POST-REBUTTAL COMMENTS: I would like to thank the authors for addressing my comments with a discussion of the suggested references, and for the additional experiments as requested by the other reviewers. I am satisfied with the rebuttals and recommend for the paper to be accepted.


Review 3

Summary and Contributions: The paper attempts to understand and explain the benefits of multi-generational self-distillation (SD). It proposes to interpret teacher-student training as amortized MAP estimation, where the teacher predictions provide instance-specific regularization benefits. This proposal is used to related label smoothing (LS) with SD. The paper also proposed a regularization technique called “Beta smoothing” and shows that SD can improve calibration. Edit: The authors tried addressing both of my comments. Regarding the first comment, I am not convinced with the explanation of why regularizing PU is not enough. Regarding the second point about extra experiments, the authors mention that they will add ablations and extra experiments (on other domains as well). Kudos to the authors but I am not sure if this approach would work out-of-the box on NLP domain and I would like to see some validation of the that. At this point. I am not changing my scores.

Strengths: * The paper does a good job of designing several small hypotheses, validating them, and using them to formulate more significant hypotheses. The experiments flow naturally with the conclusion and do not appear to be some "after-thought" to justify the conclusions. * It shows how SD can be interpreted as providing instance-specific regularization and builds on this observation to propose the MAP perspective for SD. In line 178, the authors rightly noted that this framework could be used for other problems as well. * The idea of using Beta Smoothing Labels (based on simple exponential moving average) is a useful contribution of the paper. * The paper clearly outlines the implementation choices they made and provide a reason when they deviate from the convention. (An example is Line 89), and clearly describe when the proposed theory does not yield benefits in practice (line 206, 244).

Weaknesses: * It will help if the paper describes the possible alternate formulations for Confidence Diversity (CD). I am not asking for additional experiments here. It is difficult to what extra information is CD capturing on top of Predictive Uncertainty in the current form. Or why is entropy not a good measure of "amount of spreading of teacher predictions over the probability simplex among different (training) samples" (line 115). Note that line 113 did not clarify it for me. * I think the experimental section is somewhat weak. The paper considers three datasets and two models, but all the datasets are vision datasets and are small. Moreover, the baselines are sufficient. I would like to see at least a standard distillation baseline, trained with both soft and hard labels. We don't know the number of generations for the SD baseline as well. I understand the authors point out that parameter tuning is expensive. Still, they should consider ablations (and not parameter tuning) and show that their empirical gains are consistent across different parameters.

Correctness: The equations look correct. The experimental setup does not have "design" flaws, but the empirical results are weak (refer to the weakness section). In the limited experiments that are included in the main paper, the conclusions seem to hold.

Clarity: The writing is generally good, and the paper flows well. I did not have to go back to previous sections often, to connect the dots. The paper setups a hypothesis that provides some evidence and uses that to come up with a new hypothesis. I enjoyed the writing and will just re-echo my previous comment about explaining "Confidence Diversity" clearly (esp given that it is a central idea).

Relation to Prior Work: Related work seems sufficient.

Reproducibility: Yes

Additional Feedback:


Review 4

Summary and Contributions: The paper brings a few contributions to understanding distillation and label smoothing. In short: it suggests that distillation improves when instance specific confidence varies across examples; It puts forwards a new way of label smoothing, suggests a Bayesian interpretation of distillation, and empirically shows that distillation improves calibration. More discussion of the specific points: Central to the paper is a claim that accuracy in iterative self-distillation proposed in "Born Again Neural Networks" correlates with "confidence diversity" - an introduced measure of variability in confidence in the correct class across examples. This measure is contrasted with predictive uncertainty (entropy of predictions), maximized by a uniform distribution over labels. Inspired by this observation, authors emphasize importance of instance-specific target distribution as opposed to uniform smoothing. Next, authors interpret distillation as inference of a posterior distribution given a multinomial likelihood distribution and a prior distribution being Dirichlet distribution controlled by a teacher network. A new approach for smoothing is proposed (Beta - smoothing labels, related to self-training). Empirical results demonstrate distillation improves calibration, and beta smoothing improves over uniform smoothing. **After reading the response** I appreciate authors diligent response. I especially appreciate the new results which reinforce their claims, explanation of why the claim of distillation improving calibration is not trivial, clarification of why Figure 1 does not completely contradict author's claim of importance of predictive diversity.

Strengths: Overall, the paper brings in some interesting contributions to the field of distillation: - an interesting view of effect of smoothing / distillation in the lens of Bayesian inference: they are an example-specific prior distribution over class probabilities. - interesting connection drawn between an accuracy and variability in confidence across examples - showing that distillation improves calibration is very worthwile

Weaknesses: Overall, the paper has some shortcomings which significantly hinder its impact: - The view of importance of different confidence per example is interesting, but it wasn't substantiated by ablation studies. I would like to see an experiment where a term similar to Confidence diversity was added to the objective to verify if this is indeed a useful property. - As the paper stands, I am not convinced that the Bayesian view of distillation proved helpful. The Beta smoothing of the labels indeed seems to be inspired by the Bayesian view, but such methods could also be argued for through the "instance specific regularization of the logits", where cross entropy against target distribution is added as a regularization term. I guess it can be called a Bayesian lense on the same kind of techniques, but I am not sure it opens new doors. (if it does, I will appreciate authors input) - given label smoothing has been shown to improve calibration (https://arxiv.org/pdf/1906.02629.pdf), showing that distillation improves calibration seems incremental, as KD has been interpreted as type of LS: ‘Lightning the dark: towards better understanding of KD’. - lack of comparison to self-training baseline, which seems a must given the Beta-smoothing proposed method

Correctness: I found some claims from authors to not be corresponding to some of the empirical results: - "sequential distillation indeed leads to improvements" - from Figure 1, it seems that model before distillation has the same accuracy as the model after the 10 iterations. - The claim of confidence diversity being correlated with accuracy is contradicted by Figure 1. There, accuracy over generations behaves very differently from confidence diversity. - I have doubts that fixing label smoothing to epsilon=0.15 is a sufficient comparison to draw conclusions. It would be more convincing to see to see if sweeping over different values of smoothing keep the conclusions (even for two more values, e.g. alpha=0.1, alpha=0.3).

Clarity: The paper is mostly clear, but has some significant clarity issues: - Often equation 3 is cited, but actually equation 2 is referred to (e.g. line 141). - "Nevertheless, we believe that the conclusions drawn hold in general." -> why can we think the conclusions hold in general? - I didn't understand how the 'Beta smoothing labels' algorithm is using uncertainty from the model. From what I see in section 6, the Beta distribution is not parameterized by model predictions, nor are the constructed weights for the target distribution. - "Cho and Hariharan [4] conducted a thorough experimental analysis, and made several interesting observations." - what kind of analysis? what kind of observations?

Relation to Prior Work: There are multiple works analysing why distillation helps, ablating whether it is due to the top logit weight from the teacher, or due to the "dark knowledge" - confusions to other labels (https://arxiv.org/pdf/1909.11723.pdf, "Born again neural networks"). Authors do cite these works, but should better explain how the proposed effect differs from previous work. In particular, the https://arxiv.org/pdf/1909.11723.pdf is not clearly summarized as to what contributions are brought over it (e.g., how does viewing distillation as smoothing differ from the authors' Bayesian perspective on distillation? Aren't these two views merely two sides of the same coin?) The related works section seem thin. Some papers are not explained at all, e.g. "Another experimentally driven work to understand the effect of distillation was also done in the context of natural language processing [32]." Label smoothing should also be reviewed, beyond just citing the Muller et al. paper. (e.g. its connection to entropy regularization: https://arxiv.org/abs/1701.06548)

Reproducibility: Yes

Additional Feedback: - I would like to see Figure 1 for different choices of temperature T (so, instead of Figure 2, just Figure 1 for those different values of temperature). I think it'd be interesting to see this in the light of confidence diversity correlating to accuracy in Figure 2 but not in Figure 1. - how is the ranking of the model predictions used in the Beta smoothing algorithm?

[Author Response · NeurIPS 2020]

We thank all reviewers for their constructive feedback! We are encouraged they find the proposed view of **self-**
**distillation (SD)** and **label-smoothing (LS)** as MAP insightful ([R2], [R3], [R4]), that relating accuracy to **confidence**
**diversity (CD)** interesting ([R1], [R4]), that the theoretically inspired **Beta-smoothing (BS)** is useful ([R2], [R3]) and
that experiments are carefully designed to verify our hypothesis ([R2], [R3]). We are also pleased [R4] endorsed the
importance of improved calibration. We address reviewers comments below, and will incorporate all feedback.

[R1]: **Relationships between CD, MAP, SD, and LS**: We show in section 5 that SD and LS can be unified under the
framework of amortized MAP estimation (Eqn 6). The difference between SD and LS amounts to different choices
of priors. Specifically, SD corresponds to using instance-specific priors. We argue that the instance-specific prior is
crucial for better generalization, and empirically verify the claim through comparisons of LS, BS and SD (See [R4] for
further discussion). This explains why SD outperforms LS. Please refer to our response to [R3] for discussion on CD.
**BS works worse than SD**: This is because the instance-specific priors obtained through BS are not as good as those
from a pre-trained teacher model which can more accurately capture the relative uncertainties among samples. Note SD
demands more computational resources. This does not contradict our unifying view of LS and SD as MAP estimation,
and further demonstrates the importance of a good prior. **Some technical details are not described clearly**: Intuitively,
samples that are easier to learn should be assigned a more confident label. As such, both confidence and beta samples
are sorted in ascending orders so that samples with higher confidence will get more confident beta labels. Lastly, we
want to apologize for any confusions, and will improve on clarity of the paper in general.

[R2]: We appreciate positive comments, and will update relevant works and improve the presentation accordingly.

[R3]: **Alternate formulations for CD**: CD is a measure of cross-sample variability of the prediction confidence in the
ground truth label. One can alternatively compute the variance of prediction confidence. Average predictive uncertainty
(PU) is the entropy of the softmax vectors on average (Eqn 4). PU will be high if all predictions are close to uniform. A
model can have high PU, but small CD if the cross-sample variability of predictions is small. We hypothesize that, due
to the expressivity of NNs, it is insufficient to only regularize PU (e.g., via LS), as all samples can have predictions
close to the smoothed labels, thereby overfitting. Some samples are likely more representative than others, and should
be assigned soft labels with higher confidence. As such, instance-specific priors should be used so that different samples
have different soft labels. **Experimental section is somewhat weak**: We plan on adding more experiments (See our
response to [R4] for some preliminary results). We will also include additional experiments with NLP tasks; We have
experiments with standard distillation in Appendix 9.6; Only one generation is carried out for all SD experiments.

[R4]: **Ablation studies on CD**: BS serves as an ablation study to demonstrate the importance of CD. In BS, the
soft labels on average are the same as that of LS, but have variability among the labels with samples from the Beta
distribution, thereby promoting CD. Please refer to [R1] on the implementation of BS. We have previously explored
using CD directly in training, but did not obtain good results. This could have been caused by difficulty in estimating
CD accurately using a small mini-batch size of e.g. 32. Naively promoting CD during early stage of training could
also have harmed learning. **On Bayesian view of distillation**: The MAP framework provides a unifying view to
many recently proposed regularization methods, and serves as a guide for the more principled design of regularization
techniques. For instance, it offers insights on improving distillation. Specifically, from this perspective, models with
better accuracy may not be better teachers (as empirically observed in recent literature). Instead, what really matters is a
model that captures the relative uncertainty among training samples. We hope to explore further on alternative ways of
obtaining better instance-specific priors. **Distillation improves calibration seems incremental**: We agree. However,
we stress that SD only improves calibration when temperature scaling is not applied to student model during distillation,
as suggested by our MAP framework. The commonly used distillation loss will not lead to such improvements (See
Sec. 3.1). **Lack of comparison to self-training**: We will add this experiment, and have some preliminary results
with ResNet. Following the implementation of arxiv.org/abs/1703.01780, we obtain accuracy of $73.9\%/53.2\%$ with
self-training as compared to $75.2\%/55.6\%$ with BS on CIFAR100/CUB200 respectively. **Sweeping over different**
**values of smoothing**: Preliminary results on ResNet with CIFAR100 (LS: $74.0\%$ BS: $74.3\%$ SD: $75.3\%$ when $\epsilon = 0.1$
and LS: $75.2\%$ BS: $75.1\%$ SD: $75.9\%$ when $\epsilon = 0.3$) and CUB200 (LS: $50.4\%$ BS: $53.5\%$ SD: $55.5\%$ when $\epsilon = 0.1$
and LS: $56.2\%$ BS: $56.9\%$ SD: $57.3\%$ when $\epsilon = 0.3$) suggest that instance-specific priors are beneficial for different
values of smoothing. We will incorporate further experiments. **Contradicting Fig. 1**: We acknowledge the large
fluctuation in accuracy and will rephrase accordingly. Nevertheless, a comparison of Fig. 1 and 2 shows that the max
accuracy achieved in 10 generations do agree with that of predictive uncertainty (PU) and CD, supporting our hypothesis
that BAN's improvements mainly come from increased PU and CD. **Combining Fig. 1 and 2**: Good suggestion! Based
on some preliminary results (not shown due to limited space), the correlation between accuracy and CD holds. In
addition, the degree of PU also can influence results. From the MAP perspective, PU quantifies the overall confidence
level of the instance-specific priors while CD measures the variability among them. There is an optimal CD and PU for
best performance. **Related works section seem thin**: We will extensively revise the related works section. Particularly,
our paper differs from arxiv.org/abs/1909.11723 in that their work does not highlight the importance of instance-specific
regularization. We also provide a general MAP framework and a careful empirical comparison of LS and SD.

[Meta-Review · NeurIPS 2020]

This work establishes a link between self-distillation and label smoothing. The reviewers found the idea interesting and, after rebuttal, found the connection correct. However, there remained lingering concerns about the empirical performance of the proposed beta-smoothing approach. Nevertheless, three reviewers found that the other contributions of the paper would be of interest to the NeurIPS community. Therefore, I recommend accept.